

# Predicting habitat suitability and range shifts under projected climate change for two octocorals in the north-east Atlantic

Tom L. Jenkins and Jamie R. Stevens

Molecular Ecology & Evolution Group, Department of Biosciences, University of Exeter, Exeter, United Kingdom

## ABSTRACT

Species distribution models have become a valuable tool to predict the distribution of species across geographic space and time. In this study, maximum entropy models were constructed for two temperate shallow-water octocoral species, the pink sea fan (*Eunicella verrucosa*) and dead man's fingers (*Alcyonium digitatum*), to investigate and compare habitat suitability. The study area covered the north-east Atlantic from the Bay of Biscay to the British Isles and southern Norway; this area includes both the northern range of *E. verrucosa* and the middle-northern range of *A. digitatum*. The optimal models for each species showed that, overall, slope, temperature at the seafloor and wave orbital velocity were important predictors of distribution in both species. Predictions of habitat suitability showed areas of present-day (1951–2000) suitable habitat where colonies have not yet been observed, particularly for *E. verrucosa*, where areas beyond its known northern range limit were identified. Moreover, analysis with future layers (2081–2100) of temperature and oxygen concentration predicted a sizable increase in habitat suitability for *E. verrucosa* beyond these current range limits under the Representative Concentration Pathway 8.5 scenario. This suggests that projected climate change may induce a potential range expansion northward for *E. verrucosa*, although successful colonisation would also be conditional on other factors such as dispersal and interspecific competition. For *A. digitatum*, this scenario of projected climate change may result in more suitable habitat in higher latitudes, but, as with *E. verrucosa*, there is a degree of uncertainty in the model predictions. Importantly, the results from this study highlight present-day areas of high habitat suitability which, if combined with knowledge on population density, could be used to identify priority areas to enhance protection and ensure the long-term survival of these octocoral species in the region.

# INTRODUCTION

Species distribution models (SDMs), also known as environmental (or ecological) niche models (ENMs), are a class of predictive models that seek to explain how animals and

Corresponding authors
Tom L. Jenkins,
tom.l.jenkins@outlook.com
Jamie R. Stevens,
J.R.Stevens@exeter.ac.uk

plants are distributed in geographic space and time (*Guisan & Thuiller, 2005*). These models estimate the relative suitability of habitat in a geographical area for a particular species by using locations where a species is known to occur (presences), and in some models not occur (absences), and a suite of predictor variables that represent the terrain and environmental conditions of the study area (*Elith & Leathwick, 2009*). SDMs have become a frequently used tool to address questions such as: what are the key ecological drivers that determine the range of a species?; what are the levels of habitat suitability in areas currently outside the known range of a species?; and how will projected changes in climate and the environment affect the future habitat suitability of a species? (*Warren & Seifert, 2011*). From a conservation management perspective, the ability of SDMs to provide insights on these questions has led to SDMs being considered in biodiversity assessments; some of these include: identifying priority areas for restoration and/or protection; detecting and/or monitoring pathways for invasive species; and forecasting the effects of climate change on biodiversity (*Araújo et al., 2019*). SDMs can, however, be sensitive to the input data and to the methodology and parameters used to build the models, which means it is important to assess the accuracy and reliability of model predictions, particularly if they are to be applied in a conservation management setting (*Sofaer et al., 2019*). In recent times, a number of studies have been published to address these concerns which provide a set of guidelines for assembling robust SDMs to address various study objectives (*Araújo et al., 2019*; *Feng et al., 2019*; *Sillero & Barbosa, 2020*).

SDMs have been used to investigate global habitat suitability of cold-water octocorals (*Yesson et al., 2012*) and regional habitat suitability in one or a few deep-sea species (*Tong et al., 2013*; *Lauria et al., 2017*; *Burgos et al., 2020*; *Georgian et al., 2020*; *Morato et al., 2020*). These studies have revealed novel findings about which terrain and environmental factors are the most important for determining habitat suitability in the octocoral species studied and how future suitability might be affected by projected changes in climate and the environment. Such a study, however, has yet to be applied to the pink sea fan (*Eunicella verrucosa*) and dead man's fingers (*Alcyonium digitatum*), two temperate shallow-water octocoral species that are found in the north-east Atlantic. These two species are both sessile, need substrate on which to attach, require moderate-strong water movement, and are typically recorded in coastal waters, 1–50 m depth (*Budd, 2008*; *Readman & Hiscock, 2017*), although colonies have been recorded at depths down to 200 m on the continental shelf. *Eunicella verrucosa* is a colonial gorgonian and is found from the western Mediterranean and north-west Africa (southern range) to south-west England and Wales and north-west Ireland (northern range). The species is categorised as 'vulnerable' by the IUCN Red List, and it is listed as a species of principal importance in England and Wales under the NERC Act 2006, which is reflective of its national rarity and its ecological importance as a valuable habitat provider for other benthic species, particularly when colonies aggregate to form dense gorgonian 'forests' (*Pikesley et al., 2016*; *Ponti et al., 2018*; *Chimienti, 2020*). *Alcyonium digitatum*, on the other hand, is a colonial soft coral and is found from Portugal (southern range) to parts of Norway and Iceland (northern range). Moreover, in comparison to *E. verrucosa*, *A. digitatum* is ubiquitous along all British and Irish coasts (Fig. 1). This presents an interesting opportunity to
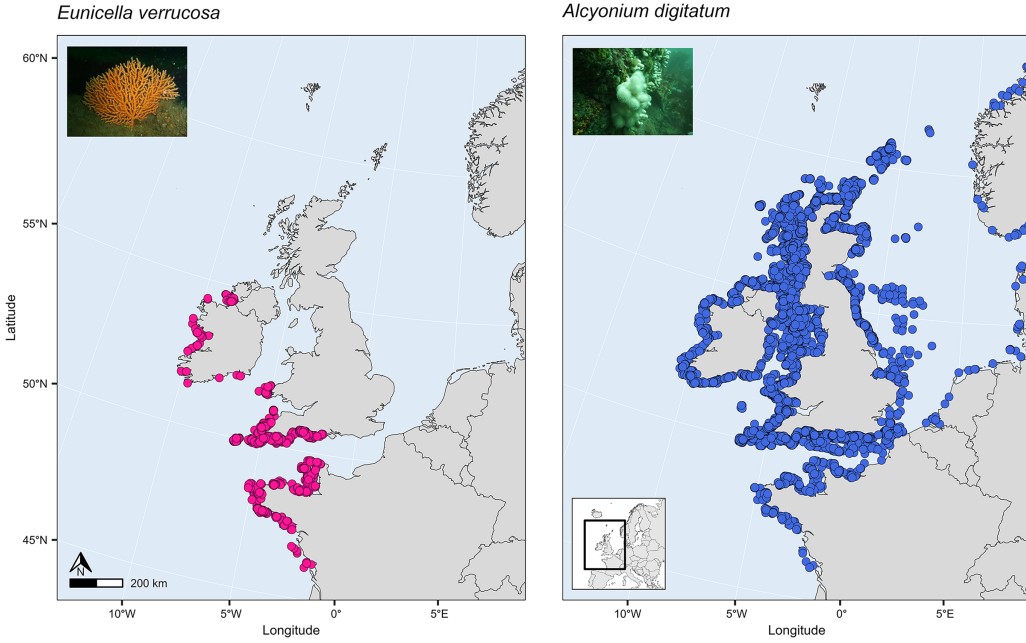

**Figure 1 Present-day distribution of the pink sea fan (*Eunicella verrucosa*) and dead man's fingers (*Alcyonium digitatum*) across the study area.** This area includes the northern range of *E. verrucosa* and the middle-northern range of *A. digitatum* in the north-east Atlantic Ocean. Presence records were collated from the Global Biodiversity Information Facility (GBIF) and from screening primary literature and reports.

explore the ecological drivers that limit the distribution of *E. verrucosa*, but apparently not *A. digitatum*, across Britain and Ireland.

The first aim in this study, therefore, was to investigate which terrain and environmental variables are the most important predictors of present-distribution in *E. verrucosa* across its northern range and in *A. digitatum* across its middle-northern range. The second aim was to predict habitat suitability across this spatial extent in both species and to potentially identify areas of suitable habitat that are currently not known to be inhabited. The final aim was to explore whether projected climate change will alter predictions of habitat suitability and potentially affect the range of either species.

# MATERIALS AND METHODS

## Study area

The study area covered all seas in north-west Europe from the Bay of Biscay to the British Isles and southern Norway (Fig. 1). This area includes the northern range of the present-day distribution of *E. verrucosa* and the middle-northern range of the present-day distribution of *A. digitatum* in the north-east Atlantic Ocean.

## Presence records

Presence-only records for both species were collated from the Global Biodiversity Information Facility (GBIF) and from screening primary literature and reports. Only unique longitude and latitude coordinates were kept, and a single record of *E. verrucosa* in
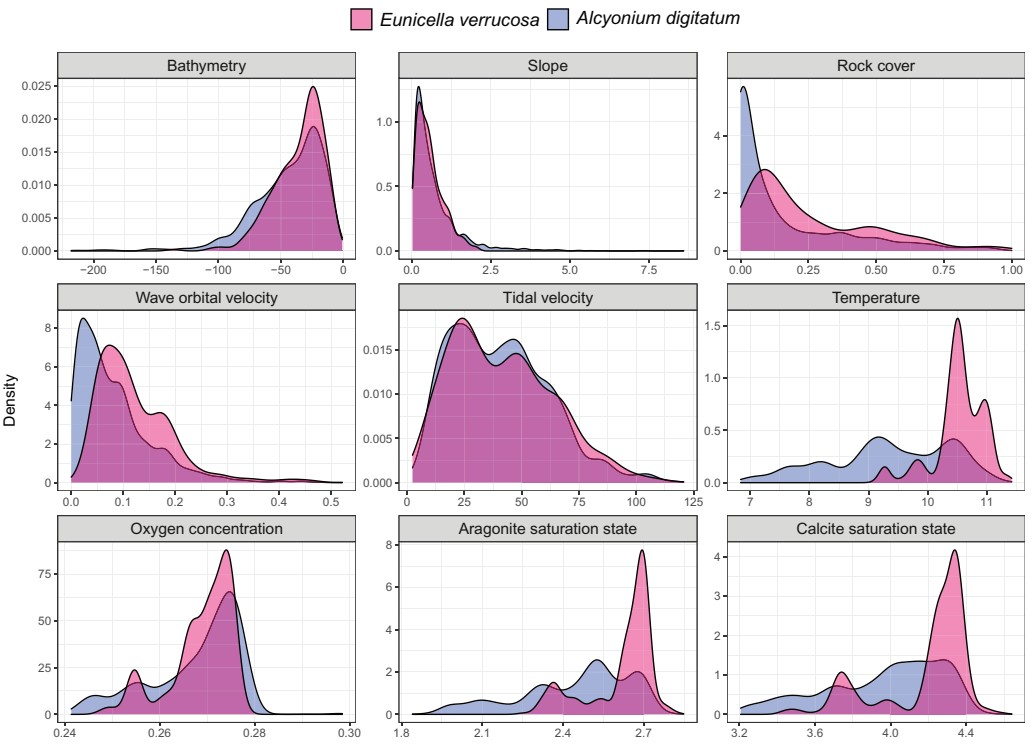

**Figure 2 Kernel density estimates for the pink sea fan (*Eunicella verrucosa*) and dead man's fingers (*Alcyonium digitatum*).** Each panel represents the kernel density estimates for a predictor variable for *E. verrucosa* (pink) and *A. digitatum* (blue) across the study area using the thinned presence points. Further details and units are presented in Table 1.

the Isle of Man (occurrence ID: 281316412) was removed because anecdotal reports suggest this observation could have been a misidentification. In addition, coordinates were removed if they were located on land. Remaining records were used to visualise the present-day distribution of *E. verrucosa* (N = 1,670) and *A. digitatum* (N = 14,460) (Fig. 1). To thin records and mitigate potential effects of survey bias prior to modelling, only one presence point per raster cell was retained and these points were further filtered to be at least 1 km apart using the R package spThin v0.2.0 (*Aiello-Lammens et al., 2015*; *R Core Team, 2021*). After accounting for missing data at some predictor variables, this left 357 presence points for *E. verrucosa* modelling and 2,373 presence points for *A. digitatum* modelling. At these thinned presence points, values for predictor variables (Table 1) were extracted to visualise an estimation of the terrain and environmental conditions experienced across the study area for each species (Fig. 2).

## Predictor variables

A set of terrain variables (static in time) and environmental variables (dynamic in time) were used as candidate predictors of present-day distribution (Table 1; Fig. S1). These variables were chosen based on their potential links to the habitat requirements of octocoral species (*Munro & Munro, 2003*; *Yesson et al., 2012*; *Holland, Jenkins & Stevens, 2017*). Bathymetry (depth) layers were downloaded from EMODnet at 0.001° resolution

**Table 1  Summary table of the terrain variables (static in time) and environmental variables (dynamic in time) used as candidate predictors in the species distribution models.**

| Predictor | Description | Unit | Source |
|---|---|---|---|
| *Static in time (terrain)* | | | |
| Bathymetry[+] | Depth of the seafloor | m | EMODnet |
| Slope | Bathymetric slope | Degrees | Computed using Bathymetry |
| Rock cover | Fraction of rock in the top 50 cm of sediment | Proportion | *Wilson et al. (2018)* |
| Wave orbital velocity | Mean wave orbital velocity at the seabed | ms$^{-1}$ | *Wilson et al. (2018)* |
| Tidal velocity | Mean tidal velocity at the seabed | ms$^{-1}$ | *Wilson et al. (2018)* |
| *Dynamic in time* | | | |
| Temperature | Temperature at seafloor | °C | *Chih-Lin et al. (2020)* |
| Oxygen | Oxygen concentration at seafloor | μmol/kg | *Chih-Lin et al. (2020)* |
| Aragonite[+] | Aragonite saturation state at seafloor | Ωar | *Chih-Lin et al. (2020)* |
| Calcite[+] | Calcite saturation state at seafloor | Ωcal | *Chih-Lin et al. (2020)* |

Notes:
The dynamic variables used in the models represented present-day values (1951–2000) and future projected values (2081–2100) for Representative Concentration Pathway (RCP) 8.5.
[+] Excluded from models (see main text for details).

and assembled into a single layer. Slope was computed from bathymetry using the terrain() function from the R package terra v1.4 (*Hijmans, 2021b*). Rock cover, wave orbital velocity and tidal velocity were downloaded from a data set assembled and published by the University of Strathclyde (*Wilson et al., 2018*). This data set covers (and is limited to) the north-west European continental shelf from the Bay of Biscay to the North Sea and the Faroe Islands at a resolution of 0.125° degrees. Temperature, oxygen concentration, aragonite saturation state and calcite saturation state at the seafloor were downloaded from a data set assembled by *Chih-Lin et al. (2020)* and cropped to the study area (~0.03° resolution). This data set represents yearly means for the periods 1951–2000 (present-day) and 2081–2100 (RCP 8.5, *i.e.*, the business-as-usual scenario); for further details on the methods used to assemble these data sets see *Morato et al. (2020)*. All rasters were projected to the Lambert Azimuthal Equal Area projection equal area grid and rescaled to match the resolution of the dynamic environmental variables using the resample() function from the R package raster v3.4-13 (*Hijmans, 2021a*) using the bilinear interpolation method. Collinearity between all predictor variables was evaluated using correlation tests and the variance inflation factor (VIF). The vifcor() function from the R package usdm v1.1-18 (*Naimi et al., 2014*) was run using all observations for each predictor variable. One variable was removed if VIF values were >10 or if the correlation was >0.70. Bathymetry was omitted as a predictor because other predictors used depth at some point in their construction and were of more interest in modelling species distribution. Calcite saturation state was originally chosen as a predictor over aragonite saturation state because calcite is the most common polymorph of calcium carbonate deposited by octocorals (*Conci, Vargas & Wörheide, 2021*), and the calcite and aragonite rasters were highly correlated across the study area. However, the use of absolute omega values in the model for *A. digitatum* resulted in virtually no suitable habitat for its current

range when projected to future layers that included calcite, which is likely an unrealistic result given that virtually all raster cells for the calcite layers used are oversaturated ($\Omega > 1$) and this is likely to be within the tolerance of both octocoral species included in this study. Therefore, as the omega values were all above one for the present-day calcite raster (100%) and were virtually all above one for the future calcite raster (99.8%), the calcite variable was omitted from the models. The final set of predictor variables comprised slope, rock cover, wave orbital velocity, tidal velocity, temperature and oxygen concentration (Table 1).

## Background locations

Reliable absence data that detail accurately where a species is not found were not available for either octocoral species. Therefore, SDMs were constructed using presence-only data and background locations in the study area. These types of models attempt to quantify statistical relationships among predictor variables at locations where a species has been observed *vs* locations where presence or absence is unmeasured within the study area (*Merow, Smith & Silander, 2013*). For each species, random selection of background points was constrained to the maximum depth of presence points and to a 50 km buffer distance drawn around presence points. The rationale here was to alleviate the influence of sampling bias by selecting more background data closer to the presence points, which are typically skewed towards near-shore/shallower sites. To reflect the differences in the number of presence points used for modelling, the total number of random background points sampled for *E. verrucosa* was 10,000 and for *A. digitatum* was 60,000.

## Maximum entropy modelling

Maximum entropy (Maxent), a machine-learning algorithm, was used to build SDMs (*Phillips, Anderson & Schapire, 2006*; *Phillips et al., 2017*). Maxent models were fitted to the data using the ENMevaluate() function from the R package ENMeval v2.0.3 (*Muscarella et al., 2014*; *Kass et al., 2021*). This function builds SDMs iteratively across a range of user-defined tuning settings, enabling different models to be compared and the optimal model to be selected. The Maxent algorithm (maxent.jar v3.4.4) from the R package dismo v1.3-5 (*Hijmans et al., 2020*) was used to construct the models. To assess model performance, the location data (presence and background points) were divided up into training and test sets using 4-fold cross-validation, and metrics were calculated for each model. The 'block' spatial partition method was used for the division, which splits the location data into four spatial groups of equal numbers (or as close as possible) that correspond to lines of latitude. The assignment of points into groups based on spatial rules attempts to mitigate spatial autocorrelation, which can overinflate model performance (*Veloz, 2009*; *Wenger & Olden, 2012*; *Roberts et al., 2017*), between points that are included in the training and validation sets. The Maxent algorithm allows the user to implement different types of models by specifying different combinations of feature classes and regularisation multipliers (*Elith et al., 2011*; *Merow, Smith & Silander, 2013*). The feature class determines the shape of the response curves, while the regularisation multiplier determines the penalty associated with adding more parameters to the model.
For example, a linear feature class will likely result in a simpler model than a model that

allows both linear, quadradic and hinge feature classes. Higher regularisation multipliers, on the other hand, impose a stronger penalty on model complexity and therefore result in smoother response curves and simpler models.

In this study, a combination of feature classes (linear, quadratic and hinge) and regularisation multipliers (1–5) were implemented using Maxent. To assess model performance, the area under the receiver operating characteristic curve (AUC) and the continuous Boyce index (*Hirzel et al., 2006*) were calculated for each model and averaged across the test sets for each combination of feature class and regularisation multiplier. $AUC_{TEST}$ values close to one indicate excellent predictive performance, while values of 0.5 indicate that the model is no better than random. The continuous Boyce index ranges from −1 to 1; positive values indicate model predictions are consistent with the distribution of presences in the evaluation data set, while values close to zero or negative values indicate that the model is no better than a random model or that the model is incorrect, respectively (*Boyce et al., 2002*; *Hirzel et al., 2006*). To select a model with the optimal fit to the data, the Akaike information criterion with a correction for small sample sizes (AICc) was computed and the model with the lowest AICc was selected as the optimal model. In addition, a null model was run and compared to the optimal model to test whether this (empirical) model differs from that of a null model. The null model was configured with the ENMnulls() function from ENMeval using the same withheld presence data and method of spatial partition as the optimal model, which enabled direct comparisons between the performance metrics of the null and the optimal model (*Bohl, Kass & Anderson, 2019*; *Kass et al., 2020*).

The percent contribution and the permutation importance of each predictor variable was extracted from the results for the optimal model. The percent contribution is a measure of which variables are contributing to fitting the model during training of the Maxent model (*Phillips, Anderson & Schapire, 2006*). The permutation importance, on the other hand, is a measure of the final Maxent model in which values of a variable are randomly permuted among the training presence and background points and the resulting value of the training AUC is recorded; a large decrease in the training AUC suggests that the model is highly dependent on that particular variable (*Phillips, Anderson & Schapire, 2006*). The predictions, interpreted as indices of habitat suitability, were then extracted and visualised using the R package tmap v3.3-2 (*Tennekes, 2018*). The optimal model was projected to future layers (2081–2100) of temperature and oxygen concentration to visualise future habitat suitability for a Representative Concentration Pathway (RCP) of 8.5, which represents a scenario of increasing emissions over time leading to high greenhouse-gas concentration levels.

## RESULTS

### Model evaluation

The hinge feature class with a regularisation multiplier of one was the optimal model for *E. verrucosa* while the hinge feature class and a regularisation multiplier of two was the optimal model for *A. digitatum* (Fig. S2). On average there was good predictive performance for *E. verrucosa* ($AUC_{TEST}$ = 0.87 ± 0.09; Boyce = 0.90 ± 0.08) and for

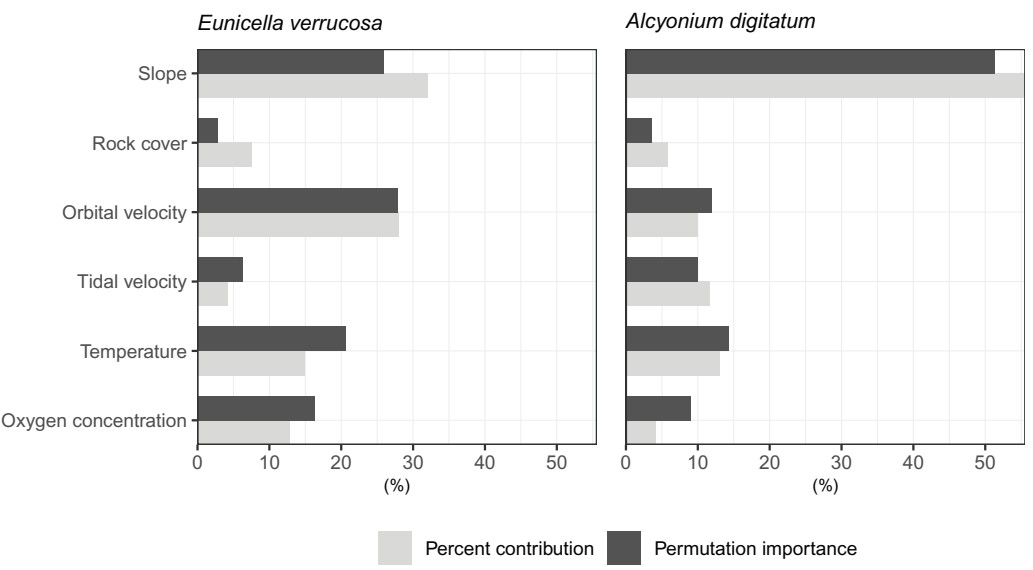

**Figure 3 Importance of each predictor variable to fitting the optimal Maxent model.** Percentages for percent contribution and permutation importance are shown for the pink sea fan (*Eunicella verrucosa*) and dead man's fingers (*Alcyonium digitatum*).

*A. digitatum* ($AUC_{TEST}$ = 0.80 ± 0.09; Boyce = 0.95 ± 0.05). Comparison tests showed strong evidence that the optimal model performed better than a null model for *E. verrucosa* ($AUC_{TEST}$ = 0.65, $p < 0.001$; Boyce = 0.29, $p < 0.001$) and for *A. digitatum* ($AUC_{TEST}$ = 0.63, $p < 0.001$; Boyce = 0.16, $p < 0.001$).

## Contribution of predictor variables

The relative contribution of each predictor variable to the SDMs was assessed by visualising the percent contribution and the permutation importance (Fig. 3). Among the predictors, slope showed high percent contribution and permutation importance for both *E. verrucosa* and *A. digitatum*. For *E. verrucosa*, wave orbital velocity, temperature and oxygen concentration were the next highest contributions to fitting the model, respectively, and although the pattern for temperature and oxygen concentration was similar in *A. digitatum*, the relative contribution of wave orbital velocity was less important in the model for *A. digitatum* compared to *E. verrucosa*. Instead, slope and tidal velocity made up a much larger contribution to model fit for *A. digitatum*. Rock cover, on the other hand, was relatively less important than the other variables for both species.

## Present-day predictions of habitat suitability

For the most part, higher predictions of habitat suitability were located in areas where both octocoral species are known to occur (Fig. 4). For *E. verrucosa*, for example, areas of high habitat suitability (0.8–1.0) were predicted across north-west France, the Channel Islands, and south-west England, and parts of Donegal (north-west Ireland). However, the model also predicted varying degrees of suitable habitat in parts of the eastern English Channel, the Irish Sea, including parts of Northern Ireland (notably Strangford Lough), parts of western Scotland and the Outer Hebrides, and inlets of Zeeland in the

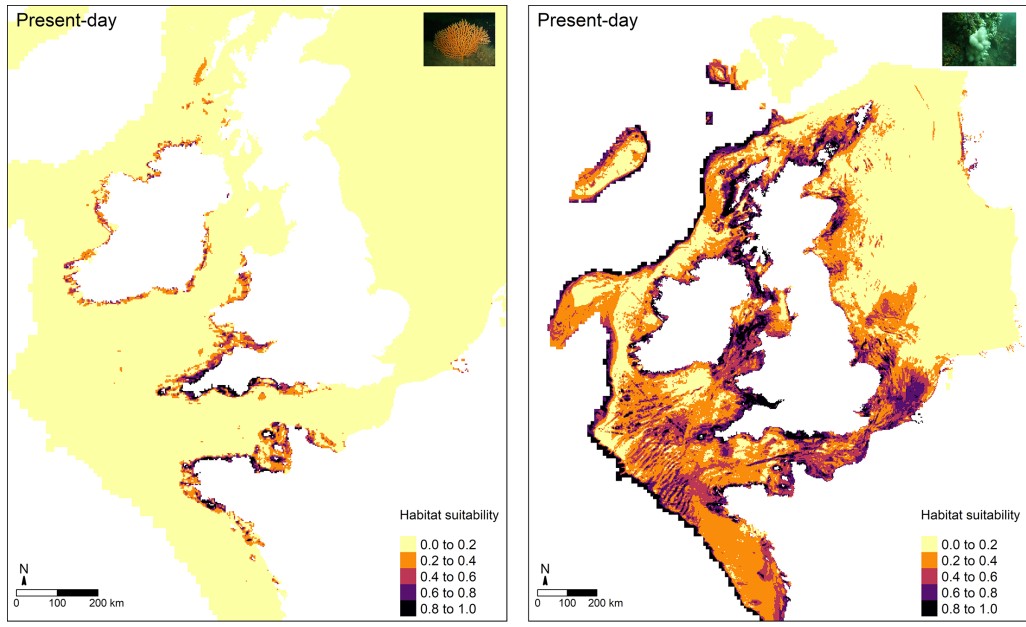

**Figure 4 Present-day predictions of habitat suitability.** Present-day predictions of habitat suitability for the pink sea fan (*Eunicella verrucosa*) and dead man's fingers (*Alcyonium digitatum*) based on the optimal Maxent model.

Netherlands, all of which have no apparent observations of *E. verrucosa*. For *A. digitatum*, high habitat suitability was predicted across both inshore and offshore areas of north-west and northern France, the Channel Islands, most of the British Isles, including the Shetland Islands and parts of the southern North Sea, parts of southern Norway, and along the Atlantic slope.

## Future predictions of habitat suitability

Future predictions (2081–2100) of habitat suitability for RCP 8.5 based on projected temperature and oxygen concentration at the seafloor showed both similarities and differences compared to the present-day predictions for *E. verrucosa* (Fig. 5). Overall, future predictions still remained high (0.8–1.0) across south-west England, the Channel Islands and north-west France. However, much higher suitability indices were apparent across Wales, Ireland, Strangford Lough, western Scotland and inlets of Zeeland. Moreover, new areas of high habitat suitability that were not observed in the present-day habitat suitability predictions were apparent in the Celtic Sea, Isle of Man, parts of Scotland, the Shetland Islands, and parts of the southern North Sea. In addition, future predictions of high habitat suitability for *E. verrucosa* covered a considerably greater spatial area than the present-day predictions.

Future predictions of habitat suitability for *A. digitatum* also revealed both similarities and differences compared to the present-day predictions (Fig. 5). Future predictions still broadly covered the same areas as the present-day predictions; however, overall, suitability predictions in the southern portion of the study area decreased, while predictions in the northern portion of the study area increased. The only areas in the southern portion not

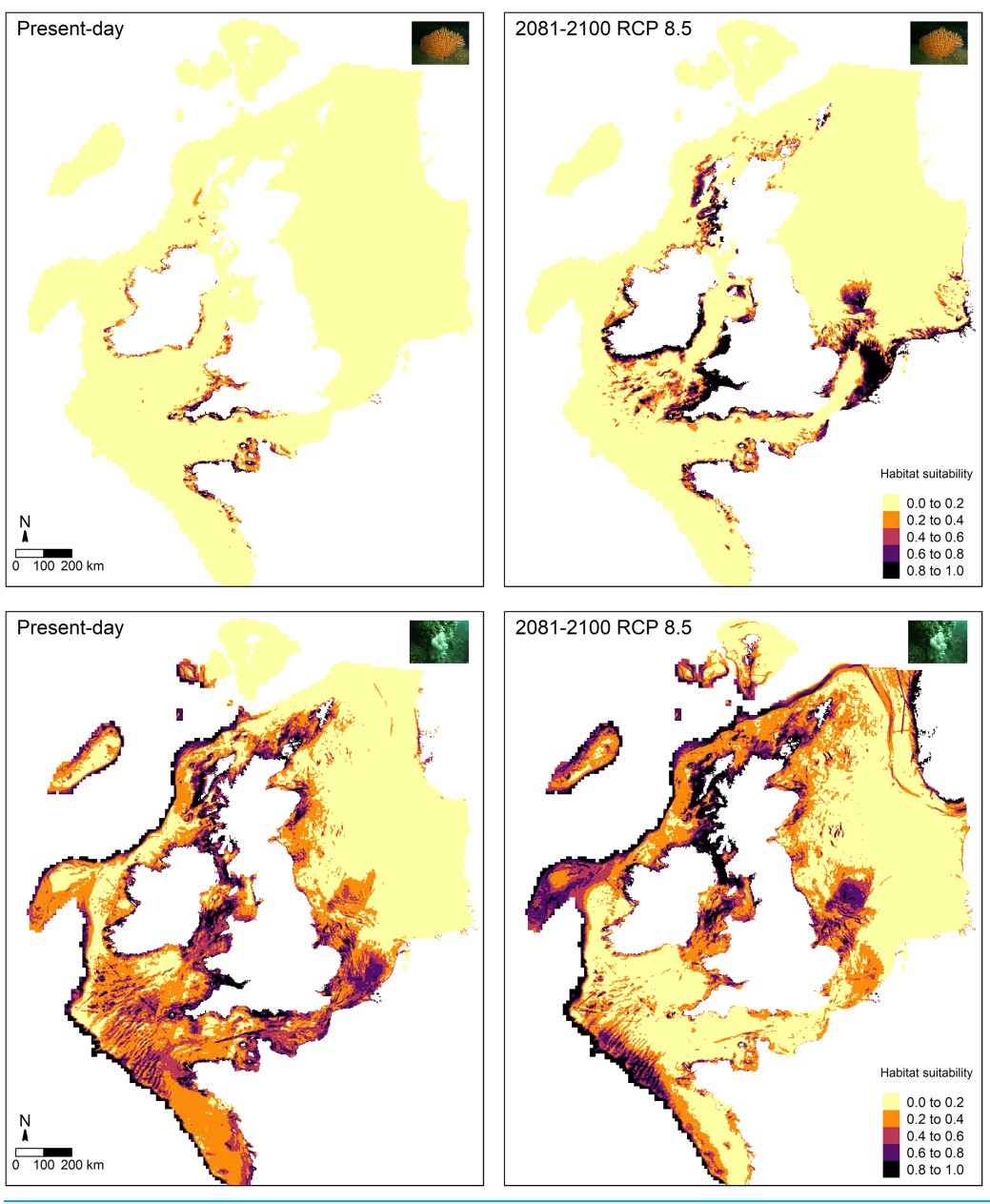

**Figure 5 Comparison of present-day and future predictions of habitat suitability.** Comparison of present-day and future predictions (2081–2100) of habitat suitability under the RCP 8.5 scenario for the pink sea fan (*Eunicella verrucosa*; top) and dead man's fingers (*Alcyonium digitatum*; bottom) based on the optimal Maxent model. Layers that were static and dynamic in time are detailed in Table 1.

affected by this pattern were areas of the Atlantic slope, where future habitat suitability remained high. Furthermore, new areas of high suitability that were not observed in the present-day predictions were apparent across fjords and coasts of Norway and across parts of the Faroe Islands.

## DISCUSSION

### Habitat preferences

Among the six predictor variables included in the SDMs, slope was particularly important for explaining the present-day species distribution of both octocoral species. This finding is not surprising since topography can influence a number of processes central to octocoral survival, including sedimentation, ocean currents and food availability (*Yesson et al., 2012*). Interestingly, wave orbital velocity was a notable predictor for *E. verrucosa* but not for *A. digitatum*. In addition, although rock cover was not as important relative to other variables in the model, it contributed slightly more to the model for *E. verrucosa* compared to *A. digitatum*, and a higher proportion of rock cover was apparent among *E. verrucosa* presence points (Fig. 2). This suggests that the ecological requirements of *E. verrucosa* may be more dependent on the availability of rock substrate and adequate current flow than *A. digitatum*. Indeed, *E. verrucosa* typically attaches to stable rock or other hard substrates *via* a basal holdfast and grows upwards and outwards towards the current flow, and while *A. digitatum* is found attached to such substrates, *A. digitatum* is also capable of settling on other substrata, including shells, cobble and other (unstable) coarse substrates (*Wood, 2013*). Tidal velocity, relative to wave orbital velocity, made only a small contribution to explaining the distribution of *E. verrucosa*. This may indicate that wave orbital velocity is more important for bringing in fresh nutrients and oxygen, both for polyps to feed on and for exporting waste products, than tidal velocity for this species.

Calcite saturation state was not used as a predictor in the final models (see methods section). Previous research has identified calcite saturation state as an important factor for determining global habitat suitability of cold-water octocorals (*Yesson et al., 2012*). Indeed, availability of calcium carbonate is critical for the formation of skeletal structures in octocoral species (*Conci, Vargas & Wörheide, 2021*). However, the calcite variable was omitted from the models because both present-day and future layers of calcite saturation state for the study area were virtually all oversaturated ($\Omega > 1$) (Fig. S1), meaning waters are supersaturated with respect to calcium carbonate and conditions, both present-day and future, are thought to be favourable for sclerite formation. Ocean acidification, an overall decrease in the pH of seawater caused by the uptake of carbon dioxide from the atmosphere (*Hoegh-Guldberg et al., 2007*), is often linked with negative effects on biomineralisation and calcification, for example, for species of scleractinian corals (*Mollica et al., 2018*), coralline algae (*Martin & Hall-Spencer, 2017*) and molluscs (*Gazeau et al., 2013*). However, octocoral species subjected to different pHs appear to show varied responses in calcification rates and sclerite structure under experimental conditions (*Rodolfo-Metalpa et al., 2015*; *Conci, Vargas & Wörheide, 2021*), which suggests that some species may have greater resilience to any potential decreases in calcite saturation state.

Temperature at the seafloor was identified as an important predictor for both *E. verrucosa* and *A. digitatum*. To our knowledge, there are no published studies that explore the thermal minima or maxima for either of these species. Therefore, the only evidence we currently have that gives insight into the thermal niches of either species are the sea temperatures at the locations where they are observed. We anticipate that the

northern range of *E. verrucosa* is constrained, at least in part, by sea surface and/or sea bottom temperature and, based on seasonal marine thermoclines (*Brown et al., 2003*), minimum winter temperature may be a candidate for limiting the distribution of *E. verrucosa* in Britain and Ireland. For *E. verrucosa* presence observations in this study, the lowest temperature extracted from the raster data set was 9.2 °C, the median temperature was 10.5 °C and the highest temperature was 11.4 °C (Fig. 2). However, this temperature data set represents averages at the seafloor (*Morato et al., 2020*) and not minimum winter temperatures, which makes it difficult to infer a preferred temperature range or thermal minima for *E. verrucosa* inhabiting the study area. Moreover, because of sampling bias (see model limitations section), colonies potentially inhabiting deeper, colder areas may have been unsampled. For *A. digitatum*, we also envisage a similar scenario whereby its northern range is limited by temperature, though the current study is less well placed to comment on this because the extent of the Strathclyde rasters (*Wilson et al., 2018*) do not cover the northern limits of the *A. digitatum* distribution (*e.g.*, northern Norway).

## Present-day habitat suitability

Maxent models built for both *E. verrucosa* and *A. digitatum* had good predictive power ($AUC_{TEST} = 0.87$ and $AUC_{TEST} = 0.80$, respectively), which suggests the models perform well at capturing areas of habitat suitability (Fig. 4). For *E. verrucosa*, the optimal model indicated potentially suitable habitat beyond its current northern range limit; these areas currently have no known presence observations in the database curated for this species. These novel areas can be broadly grouped into two categories: areas of suitable habitat in latitudes north of the range limit and in longitudes east of the range limit. The former covers: (i) parts of the Irish Sea (including Strangford Lough), (ii) northern Donegal (north-west Ireland), and (iii) parts of western Scotland; the latter covers: (i) the Isle of Wight (southern England) and parts of Normandy (northern France) in the eastern English Channel, and (ii) Zeeland (Netherlands) in the southern North Sea. Of these areas, only a few locations showed medium-high (0.6–0.8) or high (>0.8) indices of habitat suitability. For instance, there was medium-high suitability in parts of Zeeland, northern Donegal and parts of the eastern English Channel, and high habitat suitability in isolated locations off the Isle of Wight, northern Donegal, Strangford Lough and parts of north-west Normandy. In the case of Strangford Lough, this is a large marine inlet (~150 km²) on the east coast of County Down in Northern Ireland, which contains a range of marine habitats, including rocky reefs that are colonised by *A. digitatum*. It is, therefore, not unexpected to find high indices of habitat suitability for *E. verrucosa* in this inlet given the similar life histories of these two octocoral species.

There are three possible explanations for why the predictions of habitat suitability differ from the distribution of presence observations. Firstly, it is logistically and financially expensive to survey all potential areas in which *E. verrucosa* could be present across the British Isles and France, so the records database may be incomplete, and the model may have indeed captured some areas which genuinely harbour colonies of pink sea fans. Secondly, although the predictive power of the model was good, these predictions of

habitat suitability could be artefacts of the modelling process (see model limitations section). Further research, however, could explore the validity of these two explanations by conducting targeted surveys in one or more of the newly identified areas which display the highest indices of habitat suitability to confirm whether *E. verrucosa* are indeed present (or absent) at these sites. Thirdly, habitat predicted by the model may genuinely be suitable for colonisation, but *E. verrucosa* larvae may have been unsuccessful in dispersing to these sites because of low dispersal capacity and/or oceanographic barriers to dispersal (*Brown et al., 2003*). Gene flow, however, has been suggested to occur between populations in south-west Britain at distances potentially up to 480 km (*Holland, Jenkins & Stevens, 2017*), but these are estimates of genetic connectivity and do not necessarily represent a true measure of larval dispersal distances (*Lowe & Allendorf, 2010*; *Jenkins & Stevens, 2018*). Alternatively, where suitable habitat exists adjacent to *E. verrucosa* colonies, there may be extremely high interspecific competition for space and resources which prevents larvae from successfully establishing colonies at these sites and, ultimately, prevents recruitment in these areas. These two hypotheses, though, would be challenging to validate (or invalidate) *in situ*. Nevertheless, if a translocation experiment could be carried out *in situ*, researchers could test whether fragments of the nearest living colonies can survive in one of the highly suitable sites predicted by the model, where targeted surveys have confirmed the absence of *E. verrucosa* colonies.

For *A. digitatum*, the study area did not represent the northern limit of its range, meaning differences between the model predictions of habitat suitability and the distribution of presence observations were less obvious. In general, the observed patterns of habitat suitability were anticipated given that the current distribution of *A. digitatum* is ubiquitous around the British Isles and France. However, the model did show high habitat suitability on the Atlantic shelf, which currently have no presence observations in the database curated for this species. It is possible that *A. digitatum* colonies exist along the Atlantic shelf in deeper waters, but this result could also be a limitation of the modelling process as these areas represent the edge of the extent of the raster data sets used in this study (see model limitations section). The validity of these predictions could be investigated by conducting targeted surveys along the Atlantic shelf.

## Future habitat suitability under projected climate change

The Maxent model for *E. verrucosa*, projected to future layers (2081–2100) for temperature and oxygen concentration under the RCP 8.5 climate change scenario (Fig. 5), revealed some notable differences compared to the present-day predictions of habitat suitability. For instance, although suitability indices remained high around most of north-west France, the Channel Islands and south-west England, the model predicted substantial increases in habitat suitability across Wales, Ireland, western Scotland, and the southern North Sea (both inshore and offshore locations). In Zeeland (Netherlands), for example, most areas went from median-high suitability in present-day predictions to high suitability in the future predictions. Based on the raster data acquired for the RCP 8.5 scenario for the study area (Fig. 1), temperature at the seafloor is projected to increase by a median of 0.41 °C (min = −0.92, max = 2.89), while oxygen concentration at the seafloor is projected

to decrease by a median of 0.01 (min = 0, max = 0.03). This suggests that overall increases in temperature and overall decreases in oxygen concentration may indeed influence the spatial distribution of suitable habitat for *E. verrucosa*. Under this scenario of future climate change a range expansion northwards is predicted, but areas such as south-west England, Brittany (north-west France) and the Channel Islands are still predicted to remain as suitable habitat.

For *A. digitatum*, suitability indices generally decreased in the southern portion of the study area, except on the Atlantic slope, while indices in the northern portion of the study area generally increased. Under this scenario of future climate change, these predictions suggest that habitat may become more suitable in higher latitudes for *A. digitatum*. However, for both species these future predictions do not account for future changes in the static predictor variables used to build the models in this study (Table 1). Therefore, although projected changes based on the dynamic environmental variables used in this study may appear to cause range shifts under future climate change (RCP 8.5), if other variables also change, such as wave orbital velocity (possible in light of changing global weather patterns, *e.g.*, *Hu et al., 2020*; *Gupta et al., 2021*), this may affect the future predictions of habitat suitability presented here. Moreover, and importantly, these future scenarios represent change under a business-as-usual or worst-case scenario, so if mitigation steps are taken then these climate scenarios are unlikely to occur by 2100 (*Hausfather & Peters, 2020*). Nevertheless, if the dynamic variables examined here on average increase (temperature) and decrease (oxygen concentration) by some degree, then it is possible that a shift in habitat suitability, as presented for both species above, could occur towards 2100, albeit at a reduced rate and severity.

## Model limitations

There are two important limitations that may introduce uncertainty in the model predictions presented in this study. First, there may be survey bias associated with the presence records. For instance, most records of pink sea fan and dead man's fingers in the GBIF database are recorded by diver surveys which rarely go beyond depths of 30–50 m, even though both species are known to inhabit considerably deeper environments. For example, *E. verrucosa* colonies have been observed in the north-east Atlantic at depths down to 135 m (derived from data collated for this study), and *E. verrucosa* 'forests' have been recorded in the Mediterranean at depths of 65–70 m in the Ligurian Sea and at depths of 108–110 m in the Adriatic Sea (*Chimienti, 2020*). Deeper records instead come from technical diving surveys or from bottom trawling surveys where an individual is accidentally caught in the sampling gear. Nevertheless, attempts were made to control for this type of survey bias in our analysis. A related point to this is that there appears to be much greater sampling effort and records available in the GBIF database for both species from the UK, Ireland, and France. Although this may be reflective of genuine absences or rarity, this may also explain why Norway, Netherlands, Belgium, Germany, and Denmark, in particular, have very few records for *A. digitatum*. This in turn reduces the number of presence points in the model training set for these areas which, for example,

may in part explain why the predictions of habitat suitability do not entirely match the present-day distribution of *A. digitatum* in the south-eastern North Sea.

Second, there may be limitations associated with the predictor variables included in the model. For instance, one or more predictor variables could be missing from the model which are important for determining the distribution of either species. In *E. verrucosa*, a missing predictor variable could explain why some model predictions of habitat suitability extend beyond the present-day northern range limit, or potentially why in both species some areas have well supported presence records with very low indices of habitat suitability. For example, both species are commonly found on shipwrecks, which may have come to rest on an area of seabed that is otherwise unsuitable substrate for octocoral colonisation, such as bare sand or mud. These artificial reef-like habitats, for example, HMS Scylla in Whitsand Bay, Cornwall, which lies on a seabed of sand and fine shale, can often support dense, localised populations of *E. verrucosa* and *A. digitatum* (*Hiscock et al., 2010*). Therefore, obtaining accurate species data for wrecks and other artificial structures may improve the performance of the models. In addition to missing predictors, the resolution of the existing raster predictor variables used (*e.g.*, rock cover) may also explain why some areas have presence records with very low habitat suitability indices. For instance, some presence records that were located near the shoreline or within estuaries/inlets fell outside of the extent of the raster data and were subsequently removed due to missing data; this meant that the terrain and environmental conditions for this location were not used to train the model and therefore no predictions of habitat suitability were generated for these locations. This limitation could be overcome in future research by acquiring higher resolution marine terrain and environmental data sets when (if) they become available.

## CONCLUSION

This study built Maxent species distribution models for two temperate shallow-water octocoral species in the north-east Atlantic, the pink sea fan (*E. verrucosa*) and dead man's fingers (*A. digitatum*). Overall, slope, temperature and wave orbital velocity were important predictors of distribution in both species. The optimal model predictions showed areas of potentially suitable habitat where colonies have not been observed, especially for *E. verrucosa*, where areas beyond its known northern range limit were identified. Moreover, analysis with future layers (2081–2100) of temperature and oxygen concentration predicted changes in habitat suitability and potential range expansion/shifts in both species under the RCP 8.5 scenario of projected climate change.

From a conservation management perspective, these results highlight areas of high habitat suitability which could be considered in monitoring and assessment programmes. For example, enhanced protection may be afforded to areas where very high habitat suitability accords with confirmed records of high-density populations of *E. verrucosa* (gorgonian 'forests'), as found in several areas around the coasts of Devon and Cornwall in south-west England (*Pikesley et al., 2016*). Indeed, *E. verrucosa* 'forests' are thought to provide valuable habitat and shelter for a number of sessile and mobile species but are under threat from bottom-towed gear and moorings. In addition, colonies of *E. verrucosa*

are known to act as an ecological indicator of environmental condition (*Pikesley et al., 2016*) because detachment, disease and mortality may be linked to human disturbance. In conclusion, the results of this study report predictions of habitat suitability for the pink sea fan and dead man's fingers and provide insights into how populations of these octocoral species may respond to projected changes in climate and environmental conditions.

### Funding
The authors received no funding for this work.

### Competing Interests
The authors declare that they have no competing interests.

### Author Contributions
- Tom L. Jenkins conceived and designed the experiments, performed the experiments, analyzed the data, prepared figures and/or tables, authored or reviewed drafts of the paper, and approved the final draft.
- Jamie R. Stevens conceived and designed the experiments, authored or reviewed drafts of the paper, and approved the final draft.

### Data Availability
   The data and code are available at GitHub: https://github.com/Tom-Jenkins/seafan_sdm.

### Supplemental Information
Supplemental information for this article can be found online at http://dx.doi.org/10.7717/peerj.13509#supplemental-information.

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
