# Peer review of "Predicting habitat suitability and range shifts under projected climate change for two octocorals in the north-east Atlantic"

_PeerJ, doi:10.7717/peerj.13509_

## Round 0.1 · original submission · Major Revisions

Two expert reviewers have evaluated your manuscript and their comments can be seen below and in an attached PDF. Please ensure that you address all of their comments and suggestions in a revised version of the manuscript and pay particular attention to justifying the use of the 0.5 threshold cutoff.

·

Basic reporting

no comment

Experimental design

Your models use environmental data from a lat/long grid. Given the latitudinal range of your study the pixels in the south have >50% larger area than the pixels in the north. We know that variables like slope (which you are using) are sensitive to the area used in their measurement and that this affects their use in SDMs (e.g. Dolan et al https://doi.org/10.1016/j.dsr.2008.06.010). This should at least be mentioned in the discussion, but better be rectified by using equal area grids for your modelling.

Background sampling - there are several issues with the background sampling that I'd like to see addressed. You provide depth limits for the target species in the intro (50m), but your background sampling doesn't filter areas by depth. However there is a de-facto depth limit introduced by filtering to your substrate layer, which you should acknowledge. Your study area includes many areas much deeper than 50m, and table 1 reports both species seen below 100m. Including these area as background, which are wildly different from your samples, will artificially inflate your evaluation stats (see the modelling standard papers you cite in the intro). While substrate restriction is on the surface a perfectly reasonable assumption to make, it doesn't seem to have restricted your background data beyond the shelf. I'd like to see this tested by your data. What % of your presences are in the least rocky area? I'd guess there are a lot of Alcyonium that come out as being on a different substrate. If the data do not support this assumption (i.e. a significant % of observations are outside your rocky ground layer) then it seems inappropriate to use this as a background filter.

Also the rock filter does nothing to address the inherent sampling bias in your data, which is skewed to shallower, near-shore sites. There are some pretty standard methods that could adress this, e.g. selecting more background data closer to your presence points (e.g. https://onlinelibrary.wiley.com/doi/full/10.1111/ecog.04503). Given the spread of data a simple distance filter for the background data would probably cover all bases here.

Validity of the findings

Prediction maps

There are a cluster of presence points in the central north sea which the model fails to pick up. I'm guessing this is because you have chosen a 0.5 cutoff for your display. I would rather see a wider range of suitability indices displayed. There is no justification for the 0.5 threshold and I think its more useful to see the lower value predictions. Similarly for the future predictions this 0.5 threshold selection is restrictive in allowing the reader to interpret the results. There are a variety of methods for selecting a cut-off threshold (i.e. maximum sensitivity and specificity or based on percentile of presence points predicted - see maxent manual), any of these options would be better than the arbitrary selection of 0.5 which seems to be mistakenly intrepretted as a probability of presence. This is really important for assessment of future predictions.

Additional comments

Intro

lines 76-77 unnecessary repeat of distribution of E. verrucosa
lines 150-2 - Does this mean all other variables were kept at present-day values? If so please state this in the methods.

Results

line 253 - your suitability index is *not* a probability of presence and should not be called this. Reference to this should be removed everywhere in the manuscript.
line 238 - its not surprising that depth is the biggest predictor given that the majority of your samples are inshore & shallow, and most of your background points are from much deeper water.

Discussion

line 302-7 - I think you should speak about the observations of A. digitatum on low % rocky areas - it looks like there are lots of obs in very low % areas.

The critical factor for calcite (and aragonite) is the saturation state (omega). The calcite saturation horizon in the N. Atlantic is several thousand metres (e.g. feely et al https://www.pmel.noaa.gov/pubs/outstand/feel2633/feel2633.shtml). No predictions of future saturation state (that i'm aware of) have suggested that NE Atlantic will undersaturated at 50m (or anywhere close) in the near future. Also, its quite difficult to talk about future impact when your future scenarios don't include any change in saturation state (despite these layers being available and used in other, similar studies).

Can you show the thermoclines on your figures? You could easily generate these from your environmental layers. I think this would support your point that northern limit of E. verrucosa is linked with thermocline.

line 349 - I'm not convinced that a single minimum observation is a robust temperature minimum value. One outlier observation could skew this number. I think its better to present a 99th percentile value. Also this is the minimum observed annual mean temperature not the minimum bottom temperature experienced by the corals. This is an important distinction.

Figures

I find the blue dots difficult to distinguish, particularly in the scandanavian fjords, please consider changing the colour or make the country border-lines more faint. Also the scale is incorrect for the majority of latitudes on display, please reproject to an equal area grid or remove the scale bar.

Fig 4. I'm not sure what you mean by "the model was then re-run to generate future predictions". Surely the model trained on the original data was projected into your future climate layers (all but 1 of which was identical to the original layers).

Table 1 - While the min/max values observed are interesting, it would be good to show the spread of this data (e.g. include a median)

·

Basic reporting

Jenkins and Stevens report on a regional species distribution modeling approach for two species of subtidal temperate octocoral species. For the vast majority of the document, the writing is clear and the figures are well drawn (but could be slightly revised to improve engagement). The analyses appear well done and well reported (in general) but there are some areas for clarification.

Experimental design

There are some queries that I have regarding the experimental design, which are embedded as comments on the attached PDF. Generally, these are not major issues, but need some consideration.

Validity of the findings

Appear valid and insightful.

---

## Round 0.2 · Minor Revisions

I have received an evaluation of your manuscript and agree with the reviewer that the projection of the distribution of Alcyonium digitatum should be clarified by comparing the model with and without the calcite variable. Please take into account this suggestion. I look forward to receiving your resubmission.

·

Basic reporting

The authors have taken on board the majority of comments made by both reviewers. The methodology is now much more robust. This has resulted in the re-running of the models using significantly different datasets and unsurprisingly the results have changed too.

Experimental design

no comment

Validity of the findings

The only question that remains for me is the future projection for A. digitatum. The model suggests that NONE of the current range of A. dig will be suitable. This appears to be driven by the dramatic difference in Calcite saturation state exhibited by the underlying environmental layer. I understand this result is driven by the underlying data that the authors cannot control, but I think its a bit of a problem for the credibility of this study, and will very likely be over-interpreted by some readers (I could easily see a headline off of this saying beloved UK coral doomed). I'd recommend the authors do a little bit more to interpret these results and make sure this isn't the takeaway.

I would be inclined to present a model without the calcite variable to examine the relative influence of that variable on these predictions. I'd also discuss whether the real limitation is the threshold of positive or negative saturation state rather than the absolute value (you could treat the calcite layer as a binary under/oversaturated grid). It appears that the state goes from ~+4 everywhere on the shelf to ~+2. Clearly the predicted state is outside the observed range, and this is why it gets low suitability in the future projection, but omega >1 still represents an oversaturated state that is likely within Alcyonium tolerance.

---

## Round 0.3 · accepted · Accept

Thank you for taking into consideration the reviewer´s comment on the calcite variable. I am satisfied with the changes that have been made to the manuscript.